# Comparative Decoding of Physicochemical and Flavor Profiles of Coffee Prepared by High-Pressure Carbon Dioxide, Ice Drip, and Traditional Cold Brew

**DOI:** 10.3390/foods14162840

**Published:** 2025-08-16

**Authors:** Zihang Wang, Yixuan Zhou, Yinquan Zong, Jihong Wu, Fei Lao

**Affiliations:** National Engineering Research Center for Fruit and Vegetable Processing, Key Laboratory of Fruit and Vegetable Processing, Ministry of Agriculture and Rural Affairs, Beijing Key Laboratory for Food Non-Thermal Processing, College of Food Science and Nutritional Engineering, China Agricultural University, Beijing 100083, China; 2021309080501@cau.edu.cn (Z.W.); zhouyixuan1019@outlook.com (Y.Z.); zongyinquan@outlook.com (Y.Z.); wjhcau@hotmail.com (J.W.)

**Keywords:** cold brew coffee, high-pressure carbon dioxide, HPLC, GC-MS, OPLS-DA

## Abstract

High-pressure carbon dioxide (HPCD) has been widely used in the extraction of high-quality bioactive compounds. The flavor profiles of cold brew coffee (CBC) prepared by HPCD, traditional cold brew (TCB), and ice drip (ID) were comprehensively evaluated by chromatographic approaches, and their variations were investigated by multivariate statistical methods. ID produced the lightest coffee color while HPCD produced the darkest. No significant difference was found in pH among the three coffee processes. The concentrations of chlorogenic acids and caffeine were the highest in ID but the lowest in HPCD. Seventeen of the forty-eight volatiles were identified as key aroma compounds, contributing nutty, cocoa, caramel, baked, and other coffee flavors to all CBCs. Among them, linalool (OAV = 100.50) was found only in ID and provided ID with unique floral and fruity notes; 2-methyl-5-propylpyrazine (OAV = 17.70) was found only in TCB and gave a roasted aroma. With significantly lower levels of medicine-like and plastic off-flavors, HPCD had a refined aroma experience featuring nutty, cocoa, and caramel notes, though their contents were not the highest. Orthogonal partial least squares discriminant analysis (OPLS-DA) identified 36 aromas that could differentiate three cold brew methods, with TCB and HPCD being the most similar. Aroma sensory tests showed that no significant difference was perceived between TCB and HPCD. These findings provide a profound understanding of CBC flavor produced by cold brew methods from the aspect of composition, indicating that HPCD has great potential to realize TCB-like flavor characteristics in a shorter time.

## 1. Introduction

Cold brew coffee (CBC) is a beverage obtained by extracting coffee grounds with cold water (0–15 °C) [1]. It has gained increasing consumer attention due to its unique flavor profile and smooth taste [2]. Industry reports and forecasts showed that the CBC market was valued at USD 3.16 billion in 2024 and is projected to grow from USD 3.87 billion in 2025 to USD 16.22 billion by 2032 [3]. However, common cold brew extraction methods on the market, such as ice drip and immersion (traditional cold brew), require approximately a hundred times longer preparation time compared to hot brew coffee [4]. This relatively low preparation efficiency limits the further development of the CBC industry.

Recently, various novel technologies have been explored to accelerate the CBC extraction process. Ultrasonic-assisted extraction (200 W, 1 h, room temperature, 1:18 coffee-to-water ratio) could increase the total dissolved solids (TDS) content of CBC by 6–26% compared to traditional immersion without compromising aromatic profiles [5]. Reduced pressure cycles (205 mbar, 1:14 coffee-to-water ratio, 5 min per cycle, two cycles) produced CBC with no significant differences in TDS, caffeine, acidity, and phenols compared to commercial CBC [6]. High-pressure processing (300 MPa, 30 min) showed comparable quality properties to traditional 12 h CBC in physicochemical properties, sensory characteristics, and volatile compounds [7]. These technologies have also been used for the extraction of other bioactive compounds due to their outstanding extraction efficiency, such as phenolics [8], flavonoids [9], saponins [10], polysaccharides [11], proteins [12], and anthocyanins [13].

High-pressure carbon dioxide (HPCD) is a technology that treats materials with high-pressure CO_2_ [14,15]. It has been widely applied to separate bioactive compounds from plant tissues, particularly anthocyanins [16], carotenoids [17], phenolics [18], lipids [19], and alkaloids [20]. The inert property of CO_2_ and a mild operation temperature (<60 °C) provides perfect protection for oxygen and heat-sensitive bioactive compounds; the high-pressure (1–50 MPa) treatment damages cell membranes to accelerate mass transfer; and the rapid gas decompression in the end causes cell rupture to facilitate intracellular content release, all enhancing the quality and efficiency of the HPCD extraction process [21]. Yet, the use of HPCD has not been reported in CBC preparation. Considering the superiority of HPCD extraction, it could be a promising technique to achieve fast CBC preparation and bring a favorable CBC flavor at the same time.

The objective of this study was to compare the physicochemical properties and non-volatile and volatile flavor profiles of CBC prepared by HPCD and two other commonly available methods (i.e., ice drip and traditional cold brew coffee), to investigate the potential differentiating compounds among the three methods. High-performance liquid chromatography (HPLC) was utilized to determine soluble sugars, organic acid profiles, and caffeine content in CBC. Headspace solid-phase microextraction gas chromatography–mass spectrometry (HS-SPME-GC-MS) and odor activity values (OAVs) were applied to analyze the critical aroma compounds in CBC. Multivariate statistical analysis was employed to illustrate the CBC compositional difference due to the effect of the three processes. This study aims to elucidate the coffee flavor profile variation among three cold brew methods to provide an insightful understanding of CBC flavor due to cold brew method selection from a compositional aspect.

## 2. Materials and Methods

### 2.1. Materials and Chemicals

Medium-roasted Yunnan Arabica coffee beans from the same commercial batch were purchased from Baoshan Zhongka Food, Baoshan, China. Before the experiment, the coffee beans were inspected to ensure that their appearance, size, and other characteristics were nearly uniform.

All chemicals and reagents used in this study were of HPLC grade unless specified. The GC internal standard 3-octanone was purchased from TCI Development (Shanghai, China). The solvents methanol, phosphoric acid, acetonitrile, and triethylamine were obtained from Thermo Fisher Scientific (Shanghai, China). Analytical-grade magnesium oxide and sodium chloride were purchased from Sinopharm Chemical Reagent (Beijing, China). Ethanol was obtained from Solarbio Science & Technology (Beijing, China). Purified water was obtained from China Resources C’estbon Beverage (Shenzhen, China). HPLC standards of caffeine, chlorogenic acid, fructose, sucrose, maltose, lactose, citric acid monohydrate, malic acid, tartaric acid, and adipic acid were purchased from BePure (Beijing, China), and the glucose standard was purchased from Dr. Ehrenstorfer (Augsburg, Germany). Lactic acid and fumaric acid standards were obtained from Anpel Laboratory Technologies (Shanghai, China), and succinic acid standard was purchased from Tanmo Quality Inspection Technology (Changzhou, China).

### 2.2. Preparation of Cold Brew Coffee with Different Extraction Methods

Coffee beans were ground into powder using a semi-automatic coffee grinder (E10, HERO, Beijing, China). The resulting coffee powder was sieved through 30-mesh and 60-mesh standard sieves, and the fraction retained between the sieves with a particle size of approximately 250–595 μm was collected for coffee preparation. All sieved powder was prepared and used on the same day.

#### 2.2.1. HPCD Cold Brew Coffee

A total of 15 g of sieved coffee powder was placed into a 300-mesh nylon filter bag and loaded into a pressure-resistant vessel containing 150 g of purified water, maintaining a water-to-powder ratio of 10:1. The extraction matrix was placed in the processing chamber of a high-pressure carbon dioxide reactor (CAU-HPCD-1, developed by China Agricultural University; Patent No. ZL200520132590.X [22]) before sealing the vessel. The HPCD parameter settings referred to our previous coffee patent (Patent No. CN202110842969.3) [23], with 5 MPa HPCD treatment at ambient temperature (20 ± 3 °C) for 30 min. The obtained liquid was collected and reserved for subsequent use.

#### 2.2.2. Traditional Cold Brew Coffee

Traditional cold brew (TCB) coffee was prepared as described by Cai et al. [24] with slight modifications. A total of 40 g of the sieved coffee powder were placed into the brewing cartridge of the cold brew coffee machine (LPKF-LC600, Beijing Liven Technology, Beijing, China) before 400 g of purified water was added. The mixture was left to sit in a 4 °C refrigerator for 24 h. The yielding liquid was collected and reserved for subsequent use.

#### 2.2.3. Ice Drip Coffee

Ice drip (ID) coffee was prepared according to the manufacturer’s instructions for the ice drip coffee apparatus. The dripping process was carried out at ambient temperature (20 ± 3 °C). Forty grams of the sieved coffee powder was loaded into the middle container of the cold-drip apparatus (Ice Drip Tower, HERO, Beijing, China). A mixture of 100 g edible ice and 300 g purified water was placed into the upper chamber, and the mixture was allowed to drip over the coffee powder at 2 s per drop for 6 h. The obtained colored liquid in the bottom collector was reserved for subsequent use.

### 2.3. Quality Evaluation of Cold Brew Coffee with Different Extraction Methods

#### 2.3.1. Color Analysis

The CBC samples were centrifuged at 9000 rpm and 4 °C for 5 min. The collected supernatant was placed in a quartz cuvette and measured with a UV–Vis spectrophotometer (UV1800, Shimadzu, Japan), scanning from 380 to 780 nm at 1 nm intervals [25]. The obtained spectral data were imported into ColorBySpectra software (version 2017) [26] to extract color values (L*, a*, b*), and the total color difference (ΔE) between samples was calculated according to Equation (1).(1)$$\Delta E=\sqrt{{\left(∆L^{*}\right)}^{2}+{\left(∆a^{*}\right)}^{2}+{\left(∆b*\right)}^{2}}$$

In this equation, ΔE represents the total color difference, while ΔL*, Δa*, and Δb* denote the differences between two CBC samples in terms of lightness, red–green axis, and yellow–blue axis, respectively. The obtained color data were converted into a color square for better visual comparison via https://www.colortell.com/, accessed on 17 June 2025.

#### 2.3.2. pH and Coffee Extraction

pH values of three CBC samples were determined with a benchtop pH meter (PHS 25, INESA Scientific Instrument, Shanghai, China). TDS was measured using an Abbe refractometer (DR A1, ATAGO, Tokyo, Japan). The CBCs were centrifuged at 9000 rpm and 4 °C for 5 min before TDS measurement to avoid optical interference. The CBC yield (Y) calculation was performed using the following Equation (2).(2)$$Y=\frac{x\times V}{m}\times 100\%$$

In this equation, Y is the coffee extraction yield (%), x is the total dissolved solids (TDS) of coffee content (%), V is the mass of purified water (g), and m is the mass of coffee powder (g).

#### 2.3.3. Sugar Profile

Soluble sugar profiles were determined by HPLC according to the National Standard of The People’s Republic of China GB 5009.8-2023 [27]. All CBC samples were centrifuged at 9000 rpm and 4 °C for 5 min, passed through a 0.45 μm aqueous syringe filter into sample vials, and injected into an HPLC system (LC-20AD, Shimadzu, Japan). Separation was achieved using a Waters BEH Amide column (2.5 μm, 4.6 × 150 mm; Milford, MA, USA). Chromatographic separation was performed with a mobile phase of 80% acetonitrile in water containing 0.2% triethylamine, at a flow rate of 1 mL/min, over an 18 min run time. The contents of fructose, glucose, sucrose, maltose, and lactose in CBC samples were determined by the calibration curves of these sugar standards.

#### 2.3.4. Organic Acids Profile

The determination of tartaric acid, lactic acid, malic acid, citric acid, succinic acid, fumaric acid, and glutaric acid content was performed as described in the National Standard of The People’s Republic of China GB 5009.157- 2016 [28]. CBC samples around 5 g were accurately weighed and brought to 25 mL with a volumetric flask using ultrapure water. All samples were filtered through a 0.45 μm aqueous-phase membrane before injecting into an HPLC system (LC-20AD, Shimadzu, Japan) equipped with a NanoChrom ChromCore C18 column (5 μm, 4.6 × 250 mm; NAP Analytical Technology, Suzhou, China). The HPLC conditions were as follows: mobile phase A, 0.05% phosphoric acid aqueous solution; mobile phase B, methanol; flow rate, 1 mL/min. The gradient program started at 2% B and was held for 11 min; increased to 100% B at 12 min and held until 15 min; and returned to 2% B at 16 min and maintained until 30 min.

#### 2.3.5. Caffeine

Caffeine content was determined as described in the National Standard of The People’s Republic of China GB 5009.139-2014 [29]. Each accurately weighed 5 g of CBC samples was mixed with 5 mL ultra-purified water and shaken with 0.5 g of magnesium oxide, then the mixture was allowed to stand for 30 min before sending to centrifugation to collect the supernatant. The 0.45 μm aqueous-phase membrane-filtered supernatant was then injected into an HPLC system (LC-20AD, Shimadzu, Japan) equipped with a NanoChrom ChromCore AQ C18 column (3 μm, 2.1 × 100 mm; NAP Analytical Technology, Suzhou, China) for analysis. The HPLC conditions were as follows: mobile phase ratio, water/acetonitrile = 88:12 (*v*/*v*); flow rate, 1 mL/min; and run time, 4 min.

#### 2.3.6. Chlorogenic Acids

The determination of 5-caffeoylquinic acid (5-CQA), 4-caffeoylquinic acid (4-CQA), and 3-caffeoylquinic acid (3-CQA) in coffee was performed as described in the Agricultural Industry Standard of The People’s Republic of China NY/T 3514-2019 [30]. Briefly, 0.5 g of each CBC sample was diluted to 100 mL with 0.1% phosphoric acid solution. Each diluted CBC sample solution was filtered through a 0.45 μm membrane filter and injected into an HPLC system (LC-20AD, Shimadzu, Japan) equipped with a NanoChrom ChromCore C18 column (5 μm, 4.6 × 150 mm; NAP Analytical Technology, Suzhou, China) for analysis. The HPLC conditions were as follows: mobile phase A, 0.1% phosphoric acid aqueous solution; mobile phase B, acetonitrile (35:65, *v*/*v*); flow rate, 1 mL/min; run time, 6 min.

#### 2.3.7. Volatile Composition

Volatile aroma compounds in coffee were enriched using solid-phase microextraction combined with gas chromatography–mass spectrometry, following the method of Heo et al. [31]. An identical chromatographic method has been used in previous coffee studies in our lab [24,32]. Briefly, 4 mL of coffee extract, 2.5 g of sodium chloride, and 50 μL of a 3-octanone standard solution (8.2 mg/L in methanol, as internal standard) were thoroughly mixed into a 20 mL headspace vial (ANPEL Laboratory Technologies Inc., Shanghai, China) and equilibrated at 60 °C for 10 min with agitation. Then, a 50/30 μm divinylbenzene/carboxenTM/polydimethylsiloxane SPME fiber was exposed to the headspace of the vial for 45 min at 60 °C without stirring. Finally, the fiber was obtained and placed into the GC injector at 250 °C for 5 min for thermal desorption. GC-MS analysis was performed on an Agilent 7890B gas chromatograph coupled to a 5977A series mass spectrometer (Agilent Technologies, Santa Clara, CA, USA) equipped with a 30 m × 0.25 mm × 0.25 μm DB-WAX fused-silica capillary column (Agilent Technologies, Santa Clara, CA, USA). The GC conditions were as follows: injector temperature, 250 °C; carrier gas, helium at 1 mL/min; splitless injection mode. The oven temperature program was 40 °C initially, held for 10 min; ramped at 6 °C/min to 160 °C, and held for 2 min; then ramped at 8 C/min to 240 °C, and held for 10 min. The MS conditions were electron impact ionization at 70 eV; full-scan mode over m/z 33–500; threshold, 17.0; scan interval, 1 s; ion source temperature, 230 °C; quadrupole temperature, 150 °C. Volatile compounds were identified by comparing mass spectra to the standard NIST17 library, and the compounds with a matching quality over 85% were identified. The quantification of volatile compounds was performed using peak areas normalized with 3-octanone added to each sample, as an internal standard. Odor activity values (OAVs) were calculated to assess the contribution of volatile aroma compounds to coffee flavor [33], and those with OAVs ≥ 1 were considered key odor-active compounds [34]. The relevant calculations are presented below.(3)$$C_{s}=\frac{A_{s}}{A_{i}}\times 10^{4}\times C_{i}$$(4)$${\mathrm{OAV}}_{T}=\frac{C_{T}}{{\mathrm{OT}}_{T}}$$

In this equation, C_s_ is the mass concentration of the volatile compound (µg/L); C_i_ is the mass concentration of the internal standard (µg/L); A_S_ is the peak area of the volatile compound (mAU); A_i_ is the peak area of the internal standard (mAU); OAV_T_ is the odor activity value of the volatile compound i; C_T_ is the mass concentration of component T (µg/L); and OT_T_ is the odor threshold of component T in water (µg/L).

#### 2.3.8. Aroma Sensory Test

A total of 30 CBC drinkers aged 18–21 (14 males and 16 females) were recruited to participate in the aroma sensory evaluation. Data on coffee drinking habits and aroma description ratings were collected through questionnaires. The descriptive terms for sensory attributes and preferences were initially determined based on the literature [24,32,35]. After panel discussion, external consumer reviews, and participant training, the 8 descriptors (nutty, smoky, caramel, floral, roasted potato, baked, fruity, cocoa) and their intensities in 9-point scales were finalized.

During the experiment, 9 mL of each CBC sample was placed in a 30 mL transparent cup. Samples were maintained at room temperature (20 ± 3 °C), randomly coded with three-digit numbers, and presented in random order for olfactory evaluation. Participants were asked to rate the intensity of the 8 aroma descriptors in the given CBC sample on a 9-point scale. Rest intervals of 5 min were provided between sample presentations to avoid potential sensory fatigue.

### 2.4. Data Processing and Multivariate Statistical Analysis

All chemical measurements were conducted in triplicate, with the results presented as the mean ± standard deviation of the three replicates. The significance of differences among CBCs was determined using one-way ANOVA with IBM SPSS Statistics 27.0 (IBM Corporation, Armonk, NY, USA), and the pairwise comparison was performed by the Tukey method (*p*-value < 0.05). Origin 2024 (OriginLab Corporation, Northampton, MA, USA) was used to generate the OAV distribution plot, sensory radar chart, and aroma-sensory correlation heatmap. The difference in aroma profiles among the three CBCs was first evaluated by principal component analysis (PCA) using the online tool MetaboAnalyst 6.0 (https://www.metaboanalyst.ca/, accessed on 8 June 2025) [36]. Differential volatile compounds were selected by orthogonal partial least squares discriminant analysis (OPLS-DA) and performed by SIMCA 14.1 (SIMCA Umetrics, Umeå, Sweden), with model reliability validated through 200 permutation tests. Content variation on differentiating aroma was visualized by hierarchical clustering analysis (HCA) performed by TBtools-II software [37].

## 3. Results and Discussion

### 3.1. Physicochemical Quality and Non-Volatile Compounds of Cold Brew Coffee with Different Extraction Methods

Figure 1 presents the front and top views of CBC samples (9 mL each) prepared by the three methods. The measured CIELAB color values, pH, TDS, extraction yield, sugar contents, organic acid contents, and caffeine content are shown in Table 1.

Visually, HPCD CBC showed the darkest color among all the methods, followed by TCB and ID (Figure 1). The three CBCs exhibited a trend of ID > TCB > HPCD in terms of L*, a*, b* (Table 1), with significant differences (*p* < 0.05). The total color differences between the ID–HPCD, TCB–ID, and TCB–HPCD groups were 32.69, 19.46, and 13.45, respectively, suggesting a visually noticeable color difference for untrained people [38]. HPCD CBC appeared predominantly dark, suggesting that HPCD is superior to the other two methods for water-soluble coffee pigment extraction. A similar finding was reported for purple sweet potato pigment recovery, where HPCD achieved about 80% more anthocyanin recovery than water under identical extraction conditions (60 °C, 20 min) [39]. TCB showed color similar to that reported by Cai et al. [24], who applied similar CBC preparation, and the slight variation might be attributed to different coffee bean blends and roasting degrees. The lightest color of ID suggested that the iced temperature was not good for coffee pigment extraction. Aligned with previous studies, the color of CBC was influenced by the combination effect of coffee bean variety, roasting degree, extraction temperature, extraction time, and processing techniques [1,40,41].

There were no significant differences in pH among the three methods (Table 1), indicating that the presented TCB, ID, and HPCD processes had a minimal impact on cold brew coffee acidity. These findings were consistent with previous studies reporting CBC pH values ranging from 4.85 to 5.38 [42,43,44]. Cordoba et al. [45] reported that CBC pH depended on factors such as coffee variety, brewing temperature, grind size, and extraction time.

The coffee extraction rate is a key parameter for evaluating the processing of CBC. The TCB yield of 21.0 ± 0.0% was significantly higher than that of HPCD (17.3 ± 0.6%) and ID (17.0 ± 1.0%), while no significant difference was found between HPCD and ID (Table 1). The CBC yield generally ranged from 16.08% to 20.39%, with the obtained coffee TDS varying from 0.94% to 2.04% [45,46]; our results fell within these ranges. Due to the combined effect of mild temperature, high pressure, and explosive cell breakage [21], HPCD enabled CBC to achieve a yield comparable to that of 6 h continuous ID in just 30 min. Yet, 30 min HPCD could not achieve the overnight yield of TCB. A longer HPCD treatment time is suggested if a comparable amount of coffee solid yield is requested.

Among the sugars and organic acids tested, fumaric acid was the only acid whose contents were over the quantification limit (Table 1), with the concentrations ranked TCB > ID > HPCD. The chlorogenic acids (5-CQA, 4-CQA, and 3-CQA) that present astringent and bitter tastes, as well as caffeine, which gives a bitter taste, were found in all CBCs, with their concentrations following the order of ID > TCB > HPCD (Table 1). Significant differences were found among the contents of the above-mentioned compounds in different CBCs. The concentrations of sugars, caffeine, organic and chlorogenic acids were consistent with several previous studies, with the levels of sugars (total sugar: 0.226–0.461 g/100 g) and specific organic acids (citric acid: 382–10632 mg/kg, malic acid: 160–225059 mg/kg, succinic acid: 28480 mg/kg) below the presented method quantification limits [24,44,46,47,48]. Sucrose (>10 μg/mL)was the major sugar in CBC [49]; other quantified sugars included maltose, galactose, arabinose, glucose, and fructose, all in μg/mL levels [24]. Coffee bean variety, geographical origin, roast level, grind size, extraction temperature, and duration might contribute to the sugar and acid profile variation. For example, tartaric acid was not detected in Thai CBC [48], whereas tartaric acid concentration was 1.21 ± 0.02 μg/mL in TCB Costa Rica, Colombia, and Indonesia coffee blends [24]. Tang et al. [50] found that using a higher-roasting-level bean led to lower titratable acid content in Yunnan cold brew coffee. It should be noted that HPCD coffee had the lowest recovery of fumaric acid, CQAs, and caffeine, but its TDS was not the lowest (Table 1), suggesting that HPCD tends to selectively extract less bitter/astringent compounds to give a more elegant overall taste. Since caffeine, chlorogenic acids, and fumaric acid are relatively polar, the abundance of nonpolar CO_2_ in the HPCD process lowered their partition coefficients in the extraction matrix, thereby reducing their concentrations in HPCD-treated coffee [51]. ID processes had the highest contents of CQAs and caffeine but the lowest yield (Table 1), indicating iced temperature was good for coffee bioactive compound recovery. In sum, employing the TCB, HPCD, and ID processes to produce CBCs resulted in significant differences in color, yield, fumaric acid, chlorogenic acids, and caffeine contents, while pH remained statistically unchanged. HPCD tended to extract less astringent and bitter compounds, while ID had more concentrated bioactive component recovery.

### 3.2. Volatile Flavor Quality Analysis

#### 3.2.1. Volatile Compound Identification and Their Odor Activity Values

As shown in Table 2, a total of 48 volatile compounds were identified, including 16 pyrazines, 9 furans, 8 phenols, 4 pyrroles, 2 pyridines, 2 aromatic aldehydes, 2 cyclic ketones, 1 pyranone, 1 aromatic alcohol, 1 thiophene, 1 terpene, and 1 fatty acid. Their OAVs were determined to assess their sensory contribution. In studies on CBC prepared under similar conditions, Cai et al. [24] and Shi et al. [52] detected 29 and 24 of these volatile compounds, respectively. Most of the other compounds have also been identified in French pressed and ready-to-drink coffee samples [31,53].

Among the 48 compounds identified, pyrazines, furans, and phenolic compounds accounted for 33.3%, 18.8%, and 16.7% of the total volatiles, respectively. These compounds were primarily formed during the roasting process of coffee beans [54,55]. The 3-ethyl-2,5-dimethylpyrazine and 2-ethyl-3,5-dimethylpyrazine with potato-, cocoa-, roasted-, and nut-like aromas [56] were the most abundant pyrazines in all CBCs. Their concentrations among the three cold brew methods were not significant, suggesting these core coffee aromas were very easily extracted by cold water and unaffected by processing method. On the contrary, the nutty 2,6-diethylpyrazine, 3,5-diethyl-2-methylpyrazine, and 2,3-diethylpyrazine were present at lower levels or undetectable in HPCD coffee; the roasted 2-methyl-5-propylpyrazine was detected only in TCB coffee; and the roast-potato-like 2-ethenyl-6-methylpyrazine was absent in ID coffee (Table 2). Though these pyrazines varied among processes and might have potential to be selected as process-unique indicators, their notes were very similar to the abundant 3-ethyl-2,5-dimethylpyrazine and 2-ethyl-3,5-dimethylpyrazine, indicating that these variations in pyrazines would not alter the overall coffee flavor much. Furans in CBCs showed a similar pattern to that of pyrazines. The rich furfuryl acetate, 5-methylfurfural, and furfuryl alcohol that provided sweet, fruity, and fresh fruit aromas [56] showed no significant differences across the three methods, while low-abundance furans such as difurfuryl ether, furfurylideneacetone, and 2,2′-methylenebisfuran were detected exclusively in TCB coffee, exhibiting significant differences (Table 2). These low-abundance compounds, though not altering the flavor profile much, might play important roles in the aroma refinement of the CBCs, to bring complexity of sensory enjoyment to the finished product. Phenolic compounds showed a different pattern. High-abundance phenols such as spicy 4-ethyl-2-methoxyphenol and medicine-like m-cresol differed significantly among the three methods (Table 2), with their levels in HPCD being significantly lower than those in TCB and ID. For consumers who do not like these spicy, smoky, medicine-like aromas, HPCD might be an option to produce less stimulating coffee.

To analyze the aroma contributions, compounds with OAVs greater than 1 were selected, log-transformed, and plotted to the OAV distribution profile (Figure 2). The OAV > 1 numbers in TCB, ID, and HPCD CBCs were 17, 16, and 15, respectively. The overall volatile profiles were comparable to other CBCs [24,32], and all cold brew methods shared similar key aroma profiles. Four volatiles had OAV > 100, and they were major aroma contributors to the CBCs, including 3-ethyl-2,5-dimethylpyrazine with roast potato and cocoa notes, 2-ethyl-3,5-dimethylpyrazine with a nutty aroma, guaiacol with smoky and spicy notes, and linalool with citrus and floral aromas. Linalool was found only in ID, indicating that the ice temperature performed better in maintaining linalool in the final product. In fact, the CBC flavor was affected by extraction temperature, time, pressure, and solvent, as well as the specific effects associated with the given technique [21,51]. Other key aromas that showed statistical significance among the three CBCs were 4-ethyl-2-methoxyphenol with spicy and bacon notes, which was much lower in HPCD; 2-methyl-5-propylpyrazine with a roasted note, which was only found in TCB; m-cresol with a medicine-like note, which was highest in TCB but lowest in HPCD; and 1-furfurylpyrrole with waxy and plastic notes, which was much lower in HPCD. The results provided a primary impression to differentiate the three CBCs: ID was brighter and fruitier, TCB had stronger classical roast coffee notes, and HPCD tended to selectively recover less unfavorable plant notes.

**Table 2 foods-14-02840-t002:** Analysis of volatile compounds in cold brew coffee with different extraction methods.

No	CAS	Compounds	Concentration (μg/L) ^1^			Threshold (µg/L) ^2^	OAVs			Odor ^3^
			ID	HPCD	TCB		ID	HPCD	TCB	
Pyrazine										
1	13360-65-1	3-Ethyl-2,5-dimethylpyrazine	1255.08 ± 64.15 ^a^	1038.36 ± 246.2 ^a^	1392.32 ± 45.5 ^a^	1 [57]	1255.08	1038.36	1392.32	potato, cocoa, roasted [56]
2	13925-03-6	2-Ethyl-6-methylpyrazine	638.52 ± 42.96 ^a^	854.96 ± 432.36 ^a^	859.51 ± 50.93 ^a^	30 [57]	21.28	28.50	28.65	roasted potato [56]
3	13360-64-0	2-Ethyl-5-methylpyrazine	471.8 ± 30.16 ^a^	589.32 ± 266.6 ^a^	622.41 ± 39.14 ^a^	100 [57]	4.72	5.89	6.22	coffee bean, nutty [56]
4	13925-07-0	2-Ethyl-3,5-dimethylpyrazine	436.07 ± 21.41 ^a^	345.95 ± 83.57 ^a^	457.98 ± 31.69 ^a^	1 [57]	436.07	345.95	457.98	burnt, almond, roasted, nutty, coffee [56]
5	123-32-0	2,5-Dimethylpyrazine	407.71 ± 22.12 ^a^	707.26 ± 441.48 ^a^	596.6 ± 45.71 ^a^	2600 [57]	0.16	0.27	0.23	cocoa, roasted, nuts [56]
6	109-08-0	2-Methylpyrazine	379.08 ± 18.49 ^a^	662.92 ± 428.46 ^a^	563.88 ± 39.41 ^a^	60 [57]	6.32	11.05	9.40	nutty, cocoa, roasted [56]
7	108-50-9	2,6-Dimethylpyrazine	320.61 ± 18.55 ^a^	602.45 ± 405.22 ^a^	487.18 ± 37.3 ^a^	3100 [57]	0.10	0.19	0.16	cocoa, roasted, nuts [56]
8	13067-27-1	2,6-Diethylpyrazine	265.53 ± 9.53 ^a^	182.14 ± 38.01 ^b^	270.91 ± 7.79 ^a^	6 [58]	44.25	30.36	45.15	nutty, hazelnut [56]
9	13925-00-3	Ethylpyrazine	220.11 ± 9.48 ^a^	376.9 ± 249.82 ^a^	324.05 ± 26.78 ^a^	6000 [57]	0.04	0.06	0.05	peanut, butter, musty [56]
10	23747-48-0	6,7-Dihydro-5-methyl-5H-cyclopentapyrazine	148.92 ± 8.06 ^a^	87.83 ± 24.99 ^a^	143.93 ± 7.43 ^a^	-−	−	−	−	earthy, baked potato, peanut, roasted [56]
11	13925-08-1	2-Methyl-5-vinylpyrazine	146.8 ± 6.22 ^a^	117.99 ± 28.69 ^a^	145.93 ± 8.94 ^a^	−	−	−	−	durian, vegetables [59]
12	18138-05-1	3,5-Diethyl-2-methylpyrazine	692.29 ± 41.32 ^a^	285.96 ± 108.25 ^b^	405.14 ± 281.07 ^ab^	−	−	−	−	nutty, meaty, vegetable [56]
13	5910-89-4	2,3-Dimethylpyrazine	55.43 ± 7.18 ^a^	95.88 ± 88.98 ^a^	70.65 ± 3.93 ^a^	800 [58]	0.07	0.12	0.09	nutty, roasted [57]
14	15707-24-1	2,3-Diethylpyrazine	45.84 ± 2.17 ^b^	0 ± 0 ^c^	53.46 ± 3.1 ^a^	−	−	−	−	raw, nutty, green pepper [57]
15	29461-03-8	2-Methyl-5-propylpyrazine	0 ± 0 ^b^	0 ± 0 ^b^	70.8 ± 7.09 ^a^	4 [60]	0.00	0.00	17.70	roasted [60]
16	13925-09-2	2-Ethenyl-6-methylpyrazine	0 ± 0 ^b^	129.1 ± 64.06 ^a^	169.88 ± 4.88 ^a^	−	−	−	−	nutty, hazelnut [56]
Furan										
17	623-17-6	Furfuryl acetate	2416.47 ± 127.08 ^a^	2085.52 ± 553.02 ^a^	2657.17 ± 143.7 ^a^	100 [57]	24.16	20.86	26.57	sweet, fruity, banana [56]
18	620-02-0	5-Methylfurfural	2136 ± 116.03 ^a^	2609.95 ± 1167.28 ^a^	2620.03 ± 161.96 ^a^	6000 [61]	0.36	0.43	0.44	spice, caramel, maple [56]
19	98-00-0	Furfuryl alcohol	1882.72 ± 113.78 ^a^	2582.23 ± 1384.38 ^a^	2442.94 ± 91.63 ^a^	2000 [58]	0.94	1.29	1.22	sweet, creamy, vanilla [62]
20	98-01-1	Furfural	969.22 ± 45.57 ^a^	1392.74 ± 760.99 ^a^	1251.16 ± 70.27 ^a^	3000 [57]	0.32	0.46	0.42	sweet, woody, almond [56]
21	4437-22-3	Difurfuryl ether	358.15 ± 28.81 ^a^	138.9 ± 38.32 ^c^	247.64 ± 11.94 ^b^	−	−	−	−	coffee, nutty, earthy [56]
22	1192-62-7	2-Acetylfuran	302.24 ± 14.42 ^a^	486.91 ± 294.22 ^a^	411.46 ± 32.46 ^a^	10000 [57]	0.03	0.05	0.04	sweet, balsam, almond [56]
23	1193-79-9	2-Acetyl-5-methylfuran	67.47 ± 5.68 ^a^	60.87 ± 20.3 ^a^	81.66 ± 7.82 ^a^	−	−	−	−	sweet, musty, hay, coconut, coumarin [56]
24	623-15-4	Furfurylideneacetone	64.26 ± 3.14 ^a^	0 ± 0 ^c^	24.37 ± 0.81 ^b^	−	−	−	−	balsamic, creamy, vanilla-like [56]
25	1197-40-6	2,2′-Methylenebisfuran	0 ± 0 ^c^	130.38 ± 23.39 ^b^	207.16 ± 8.97 ^a^	−	−	−	−	rich roasted [63]
Phenol										
26	2785-89-9	4-Ethyl-2-methoxyphenol	425.85 ± 42.45 ^a^	232.92 ± 45.45 ^b^	345 ± 18.89 ^a^	16 [64]	26.62	14.56	21.56	spicy, smoky, bacon [53]
27	7786-61-0	4-Hydroxy-3-methoxystyrene	318.89 ± 64.27 ^a^	234.41 ± 129.72 ^a^	366.46 ± 87.09 ^a^	19 [64]	16.78	12.34	19.29	woody, clove, amber [53]
28	90-05-1	Guaiacol	248.68 ± 19.58 ^a^	183.66 ± 64.04 ^a^	230.23 ± 22.56 ^a^	1.6 [57]	155.42	114.79	143.89	phenolic, smoke, spice [53]
29	108-95-2	Phenol	150.99 ± 20.34 ^a^	172.37 ± 54.29 ^a^	137.5 ± 4.66 ^a^	2400 [60]	0.06	0.07	0.06	phenolic, plastic rubber [56]
30	108-39-4	m-Cresol	124.72 ± 8.58 ^a^	72.42 ± 18.72 ^c^	94.2 ± 5.76 ^ab^	31 [58]	4.02	2.34	3.04	medicinal, woody, leather [56]
31	95-48-7	o-Cresol	68.08 ± 10.75 ^a^	0 ± 0 ^b^	56.97 ± 2.93 ^a^	260 [58]	0.26	0.00	0.22	musty, medicinal herbal, leathery [56]
32	620-17-7	3-Ethylphenol	36.72 ± 7.49 ^a^	0 ± 0 ^b^	0 ± 0 ^b^	1.7 [58]	21.60	0.00	0.00	leather, ink-like [64]
33	96-76-4	2,4-Di-tert-butylphenol	66.08 ± 13.11 ^a^	0 ± 0 ^b^	0 ± 0 ^b^	500 [58]	0.13	0.00	0.00	phenol [65]
Pyrrole										
34	1438-94-4	1-Furfurylpyrrole	273.19 ± 26.01 ^a^	181.3 ± 38.91 ^b^	287.42 ± 15.33 ^a^	100 [58]	2.73	1.81	2.87	plastic, waxy [56]
35	1072-83-9	2-Acetylpyrrole	275.67 ± 30.22 ^a^	249.43 ± 58.05 ^a^	320.82 ± 14.05 ^a^	−	−	−	−	musty, nut skin, maraschino [56]
36	1192-58-1	N-Methylpyrrole-2-carboxaldehyde	254.75 ± 12.78 ^a^	269.73 ± 122.71 ^a^	302.16 ± 15.16 ^a^	37 [58]	6.89	7.29	8.17	roasted, nutty [56]
37	1003-29-8	Pyrrole-2-carboxaldehyde	125.7 ± 9.21 ^a^	136.57 ± 36.1 ^a^	144.03 ± 4.87 ^a^	65000 [58]	0.00	0.00	0.00	musty, beefy, coffee [56]
Pyridine										
38	67402-83-9	1-Acetyl-1,4-dihydropyridine	0 ± 0 ^b^	0 ± 0 ^b^	117.63 ± 10.16 ^a^	170000 [58]	0.00	0.00	0.00	animal, floral, moth ball [56]
39	1122-62-9	2-Acetylpyridine	0 ± 0 ^b^	0 ± 0 ^b^	56.61 ± 7.08 ^a^	19 [58]	0.00	0.00	2.98	popcorn, corn chip, fatty, tobacco [56]
Aromatic aldehyde										
40	100-52-7	Benzaldehyde	138.82 ± 24.36 ^a^	132.25 ± 61.46 ^a^	137.51 ± 12.18 ^a^	320 [66]	0.43	0.41	0.43	cherry [66]
41	4411-89-6	2-Phenyl-2-butenal	78.55 ± 7.13 ^a^	0 ± 0 ^c^	63.66 ± 3.36 ^b^	883.8 [67]	0.09	0.00	0.07	sweet, narcissus, cortex [56]
Cyclic ketone										
42	78-59-1	Isophorone	0 ± 0 ^b^	0 ± 0 ^b^	54.13 ± 6.36 ^a^	11000 [58]	0.00	0.00	0.00	woody, sweet, green, camphoraceous [56]
43	80-71-7	Methylcyclopentenolone	45.01 ± 2 ^a^	0 ± 0 ^b^	43.11 ± 2.44 ^a^	300 [58]	0.15	0.00	0.14	caramel maple syrup [56]
Pyranone										
44	118-71-8	3-Hydroxy-2-methyl-4-pyrone	0 ± 0 ^c^	131.93 ± 34.12 ^b^	220.65 ± 28.15 ^a^	5800 [60]	0.00	0.02	0.04	sweet, caramel, cotton candy [68]
Aromatic alcohol										
45	60-12-8	Phenethyl alcohol	64.35 ± 1.43 ^a^	0 ± 0 ^b^	0 ± 0 ^b^	564.23 [69]	0.11	0.00	0.00	floral, rose [69]
Thiophene										
46	98-03-3	2-Thenaldehyde	79.03 ± 8.55 ^a^	0 ± 0 ^b^	0 ± 0 ^b^	5000 [58]	0.02	0.00	0.00	sulfurous [63]
Terpene										
47	78-70-6	(±)-Linalool	58.29 ± 19.21 ^a^	0 ± 0 ^b^	0 ± 0 ^b^	0.58 [70]	100.50	0.00	0.00	citrus, floral [70]
Fatty acid										
48	503-74-2	Isovaleric acid	114.16 ± 9.41 ^a^	122.21 ± 74.19 ^a^	126.49 ± 9.99 ^a^	33.4 [71]	3.42	3.66	3.79	cheese, dairy, sour, fatty [56]

The data presented represent mean value ± standard deviation (*n* ≥ 3). Significant differences at the *p* < 0.05 level are denoted by distinct lowercase letters within the same row (a is the largest). The odor threshold of the compound in water provided in the literature; Odor descriptions provided in the literature.

#### 3.2.2. Aroma Sensory Analysis

Figure 3 presents the radar plot of the aroma sensory evaluation of three CBCs on a 9-point scale, with the higher scores indicating more pronounced aromas. The overall smell intensities and descriptions were comparable to previous studies on CBC, with nutty, fruity, and floral notes being the major flavor characteristics [24,32]. ID had the highest scores for floral and fruity (Figure 3), which was in agreement with our compositional analysis, which identified linalool as a key aroma contributor only in ID (Table 2 and Figure 2). TCB scored highest in roasted potato notes (Figure 3), which might be associated with 2-methyl-5-propylpyrazine that featured a roasted note and was only found in TCB (Table 2 and Figure 2). HPCD scored highest in the nutty, cocoa, smoky, roasted, and caramel aromas (Figure 3). Although the concentrations of typical coffee aroma contributors were not the highest in HPCD, the results suggested that a reduction in less unfavorable notes (medicine-like m-cresol, waxy and plastic 1-furfurylpyrrole, etc.) may allow the favorable nutty, cocoa, and roasted coffee aromas to stand out, and provide a more refined sensory experience.

Sensory perception of coffee is a combination of the components of the coffee matrix, the release rate of volatile substances, and the interaction of olfactory receptors [53,56,62,72,73,74,75]. The relatively high sensory scores of HPCD were worthy of special mention. HPCD treatment has been shown to achieve significant reductions in off-flavor substances in Chinese water chestnut, such as o-Cresol and 3-ethylphenol [76], similar to the low levels of m-cresol and 1-furfurylpyrrole found in this study (Table 2). The removal of off-flavors is likely to allow participants to perceive the core pleasant aromas more distinctly, resulting in higher sensory scores. Additionally, HPCD’s impact on the coffee matrix may have contributed to this change, as the high-pressure treatment of HPCD could have altered the structure of macromolecules in coffee, thereby influencing the release kinetics of aroma molecules [21,72]. This modification might have made the aroma compounds in HPCD coffee more readily volatilize, compensating for their lower concentration in the solution.

Yet, none of the aroma sensory scores showed statistical significance among the three extractions (Appendix A), suggesting the minor aroma differences among CBCs may be sensed by very sensitive individuals, but cannot be easily recognized by common consumers.

To explore the potential correlation between the contents of active volatile compounds and aroma sensory intensities, a Pearson correlation analysis was conducted, and the correlation heatmap was plotted (Figure 4). Eight aroma descriptors related to the content of volatile substances were divided into four groups. The first group involved nutty, caramel, baked, and cocoa, which were positively correlated with 2-methylpyrazine (nutty, cocoa, roasted [56]) and furfuryl alcohol (sweet, creamy, vanilla [62]), while they were negatively correlated with 4-ethyl-2-methoxyphenol (spicy, smoky, bacon [53]), guaiacol (phenolic, smoke, spice [53]), and m-cresol (medicinal, woody, leather [56]). The second group included floral and fruity notes, with a correlation with the volatile compounds opposite to that of the first group. The third group had a roasted potato note, which demonstrated significant positive correlations with N-methylpyrrole-2-carboxaldehyde (roasted, nutty [56]), 2-ethyl-6-methylpyrazine (roasted, potato [56]), 2-ethyl-5-methylpyrazine (coffee bean, nutty [56]), and isovaleric acid (cheese, sour fatty [56]). The last group had the smoky note alone, which had no significant correlation with any key aroma compounds in CBCs in this study. In sum, the sensory results were generally corroborated by the aroma descriptors and their quantities in Table 2. Some variations might be due to the aroma proportion variation and their balance in the CBC matrices, which led to either synergistic or antagonistic effects on aroma perception [77].

### 3.3. Differentiating Volatile Compounds Among Three Extraction Methods

#### 3.3.1. Principal Component Analysis of Volatile Compounds from Different Extraction Methods

PCA was conducted to determine whether the aroma characteristics of the three CBCs could be separated. As shown in Figure 5A, the first two principal components (PC1 and PC2) accounted for 85.9% and 10.1% of the total variance, indicating the model captured the majority of the variability in the dataset [78]. The score plot (Figure 5A) provides a clear visualization of the separation among the CBCs from the three extraction methods in the PC1–PC2 space, revealing significant differences in their flavor compositions.

Figure 5B shows the corresponding PCA biplot. The biplot identified the top ten compounds contributing most to the separation: furfuryl acetate (OAV > 1), furfuryl alcohol (OAV > 1), 5-methylfurfural, 3-ethyl-2,5-dimethylpyrazine (OAV > 1), 3,5-diethyl-2-methylpyrazine, furfural, 2,5-dimethylpyrazine, 2-methylpyrazine (OAV > 1), 2,6-dimethylpyrazine (OAV > 1), and 2-ethenyl-6-methylpyrazine (OAV > 1). These compounds predominantly impart sweet, nutty, roasted, cocoa, and fruity aromas (Table 2). The loading distribution revealed that HPCD samples were enriched in pyrazines that confer nutty and cocoa notes, ID samples contained higher levels of esters that impart fruity aromas, and TCB samples were rich in furan derivatives that provide roasted and caramel-like notes. All these aroma classifications and their odor descriptions were generally aligned with our sensory results, that HPCD coffee was characterized by pronounced nutty notes, ID by fruity notes, and TCB by roasted aromas (Figure 3).

#### 3.3.2. Identification of Key Differential Volatile Compounds

OPLS-DA was introduced to screen for potential differential volatile compounds among the three CBCs. OPLS-DA is a supervised pattern-recognition multivariate statistical analysis method that integrates orthogonal signal correction with partial least squares discriminant analysis [79,80,81]. To date, OPLS-DA has been widely applied in bioinformatics, chemometrics, and food flavor chemistry [81,82,83].

The OPLS-DA models were constructed on normalized data and demonstrated clear separation between each CBC pair (Figure 6(A1,B1,C1)). The discriminative performance of the models was assessed by independent variable fit index (R^2^X), dependent variable fit index (R^2^Y), and model prediction index (Q^2^). R^2^ and Q^2^ values exceeding 0.5 usually indicate acceptable model reliability [84]. With all models having R^2^ and Q^2^ values > 0.8 (Appendix A), the models were considered to have acceptable performance. The risk of model overfitting was evaluated by permutation tests. After 200 permutation tests (Figure 6(A2,B2,C2)), all points on the right were consistently higher than those on the left, and the Q^2^ regression line *Y*-axis intercept was less than zero, suggesting that there was no overfitting of the model, and the model was valid (Appendix A) [85]. Therefore, the OPLS-DA models are reliable and suitable for differential-substance screening.

Intergroup differentiating volatile compounds were identified by a combination of OPLS-DA S-plots (Figure 6(A3,B3,C3)) and variable importance in projection (VIP) values (Table 3). Taking VIP > 1 as a criterion for screening, a total of 36 differential substances were identified. The HCA heatmap (Figure 7) was then plotted to better visualize the quantitative differences in the 36 aromas among cold brew methods. The bottom half of Figure 7 provides clear aroma quantity differences among the three CBCs, showing that the aromas that had clear differentiating power were the ones that were not detected in at least one cold brew method. Among the three methods, TCB and HPCD were the most similar, with ID being separated into another group (Figure 7). This might be associated with five aromas only being found in ID, including the fruity linalool, which contributed to the unique fruity and flora smell of ID coffee (Figure 3). Moreover, three aromas, including 2-ethenyl-6-methylpyrazine with a nutty note, 2,2′-methylenebisfuran with a rich roasted note, and 3-hydroxy-2-methyl-4-pyrone with a caramel note were both detected in TCB and HPCD (Figure 7), but not in ID, which were aligned with the sensory perception that TCB and HPCD featured stronger roasted, nutty, and caramel smells (Figure 3). There were nine compounds that were found in TCB but not in HPCD to differentiate the two, which included the unfavorable medicine-like o-cresol and woody green isophorone (Figure 7), but none of which had OAV > 1 (Table 2).

With the OPLS-DA, 36 key differential volatile compounds were screened, among which only 4-ethyl-2-methoxyphenol, m-cresol, 3-ethylphenol, 2-acetylpyridine, 1-furfurylpyrrole, n-methylpyrrole-2-carboxaldehyde, 3-ethyl-2,5-dimethylpyrazine, 2-ethyl-6-methylpyrazine, 2-ethyl-5-methylpyrazine, 2-methylpyrazine, 2,6-diethylpyrazine, 2-methyl-5-propylpyrazine, furfuryl alcohol, and (±)-linalool had OAV > 1 (Table 2). And these 14 key aroma contributors reflected the sensory experience difference that ID was fruitier, TCB was more roasted, and HPCD had more caramel, nutty, and cocoa notes. It is true that the three cold brew methods can be differentiated by compounds in trace amounts, but not all such minor compositional differences are reflected as a perceptible difference in the sensory experience.

## 4. Conclusions

This study comprehensively compared the non-volatile and volatile profiles of three CBCs. ID, TCB, and HPCD showed no significant difference in coffee pH values. Among the 48 volatiles detected, 17 aromas had OAV > 1 and were identified as key aroma contributors, including 3-ethyl-2,5-dimethylpyrazine with roast potato and cocoa notes, 2-ethyl-3,5-dimethylpyrazine with a nutty aroma, and guaiacol with smoky and spicy notes. ID (6 h) produced light-colored coffee, featuring a fruity aroma brought by linalool (OAV = 100.50) that was only found in ID, and the CQAs and caffeine contents were significantly higher than those of other cold brew methods. TCB (4 °C, 24 h) had the highest yield among all methods, featuring a classical coffee roasted aroma brought by 2-methyl-5-propylpyrazine (OAV = 17.70) and moderate CQAs and caffeine contents. HPCD (5 MPa, 30 min) extracted coffee with the darkest color and recovered the lowest concentrations of CQAs and caffeine, featuring nutty, cocoa, and caramel aromas, though their contents were not the highest. The HPCD treatment could selectively recover less unfavorable notes (medicine-like m-cresol, waxy and plastic 1-furfurylpyrrole, etc.) and facilitate the favorable coffee nutty, cocoa, and roasted aroma standing out, thus providing a more refined sensory experience.

The aroma sensory tests suggested no significant perceived differences between HPCD and TCB coffees, and the aroma profiles clustered HPCD coffee closely with TCB coffee, indicating that HPCD has great potential to serve as an efficient alternative to TCB coffee. A more sophisticated sensory test involving tasting and additional HPCD processing optimization is highly recommended to thoroughly evaluate the feasibility of HPCD in large-scale cold brew coffee production.

## Figures and Tables

**Figure 1 foods-14-02840-f001:**
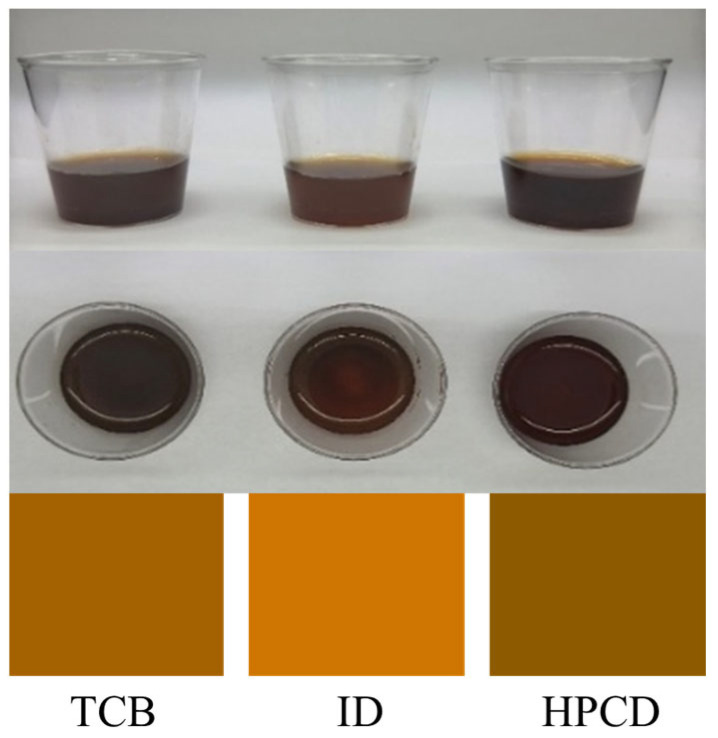
Appearance of cold brew coffee obtained by different extraction methods.

**Figure 2 foods-14-02840-f002:**
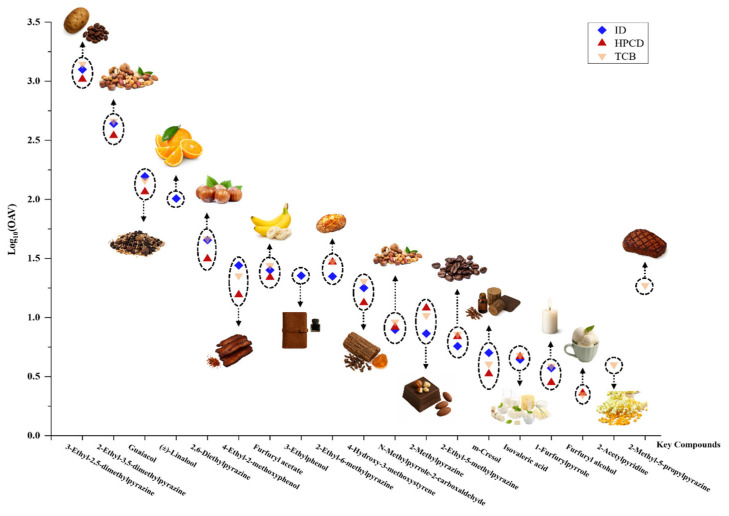
OAV distribution map in cold brew coffee with different extraction methods (OAV > 1).

**Figure 3 foods-14-02840-f003:**
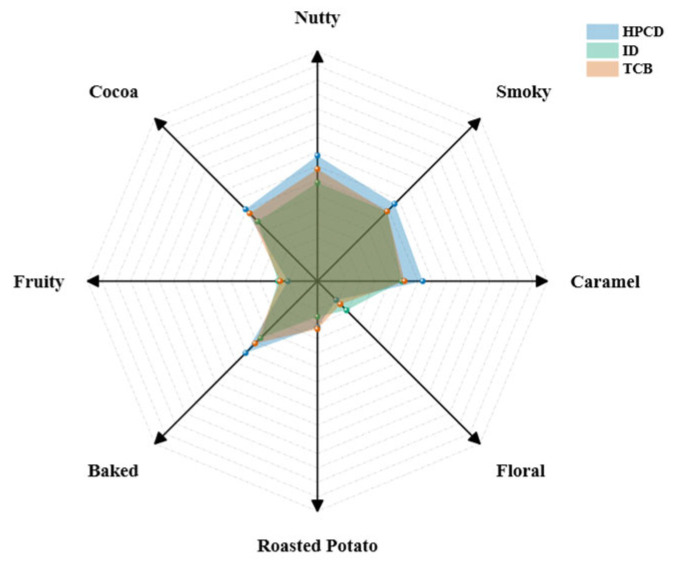
Aroma sensory radar chart with different extraction methods.

**Figure 4 foods-14-02840-f004:**
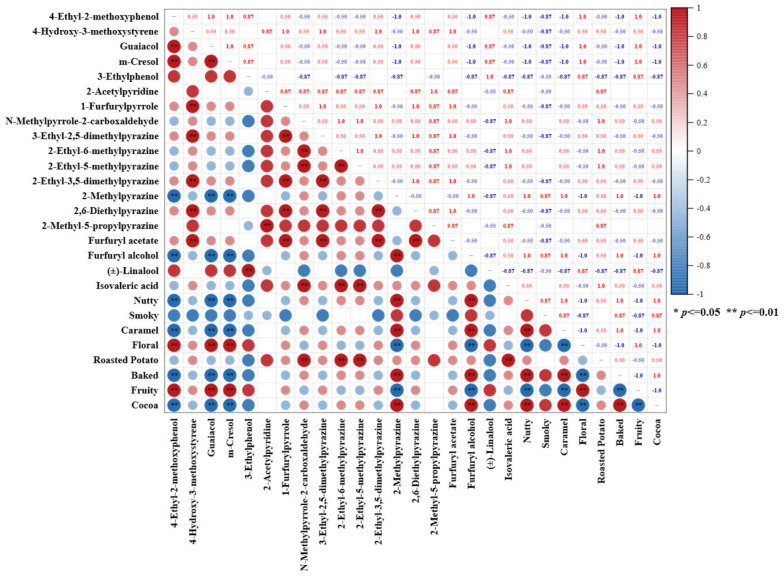
Pearson correlation plot of CBC key aroma compounds and aroma sensory scores.

**Figure 5 foods-14-02840-f005:**
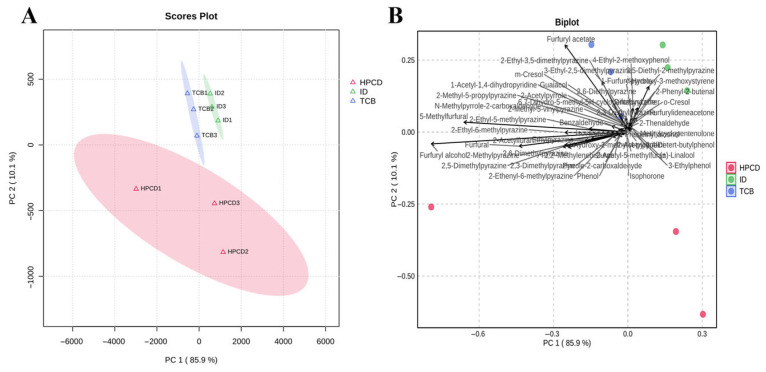
Principal component analysis plots of volatile compounds in cold brew coffee from different extraction methods, (**A**) score plot, (**B**) biplot.

**Figure 6 foods-14-02840-f006:**
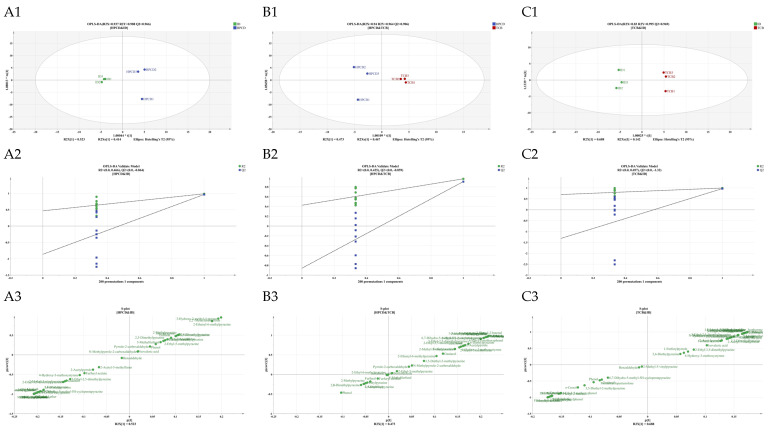
Orthogonal partial least squares discriminant analysis of volatile compounds in cold brew coffee from different extraction methods. (**A1**,**B1**,**C1**) are OPLS-DA separation for HPCD and ID, HPCD and TCB, TCB and ID; (**A2**,**B2**,**C2**) are OPLS-DA model validation plot for OPLS-DA separation for HPCD and ID, HPCD and TCB, TCB and ID; (**A3**,**B3**,**C3**) are S-plots for OPLS-DA separation for HPCD and ID, HPCD and TCB, TCB and ID.

**Figure 7 foods-14-02840-f007:**
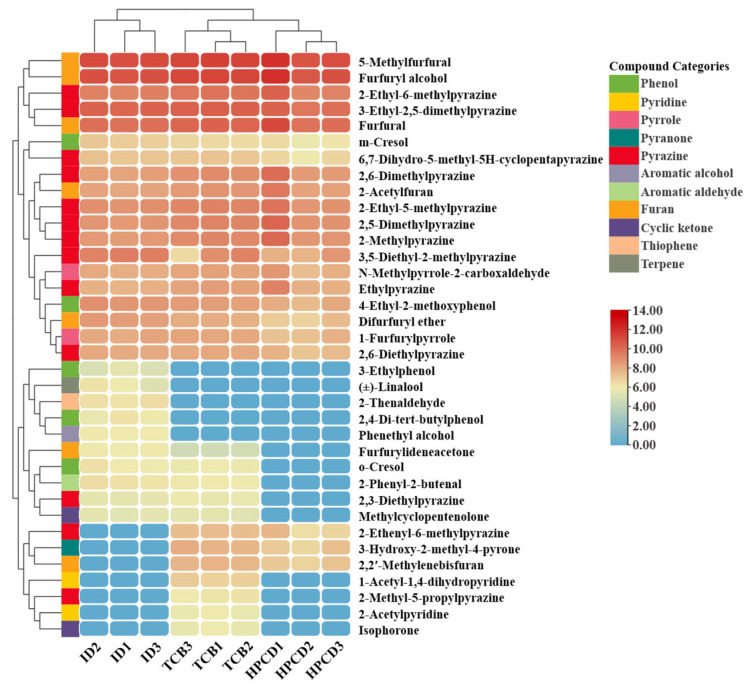
Heatmap of hierarchical clustering analysis of differentiating volatiles among three cold brew methods.

**Table 1 foods-14-02840-t001:** Physicochemical quality and partial non-volatile components of cold brew coffee under different extraction methods.

	TCB	HPCD	ID
Color L*	47.33 ± 0.19 ^b^	42.91 ± 0.56 ^c^	57.59 ± 0.00 ^a^
Color a*	21.16 ± 0.09 ^b^	14.81 ± 0.13 ^c^	27.98 ± 0.08 ^a^
Color b*	67.09 ± 0.10 ^b^	56.08 ± 0.47 ^c^	82.16 ± 0.06 ^a^
pH	5.22 ± 0.02 ^a^	5.24 ± 0.02 ^a^	5.25 ± 0.02 ^a^
TDS	2.10 ± 0.00 ^a^	1.73 ± 0.06 ^b^	1.70 ± 0.10 ^b^
Extraction rate (%)	21.0 ± 0.0 ^a^	17.3 ± 0.6 ^b^	17.0 ± 1.0 ^b^
Fructose (g/100 g)	ND (<0.5)	ND (<0.5)	ND (<0.5)
Glucose (g/100 g)	ND (<0.5)	ND (<0.5)	ND (<0.5)
Sucrose (g/100 g)	ND (<0.5)	ND (<0.5)	ND (<0.5)
Maltose (g/100 g)	ND (<0.5)	ND (<0.5)	ND (<0.5)
Lactose (g/100 g)	ND (<0.5)	ND (<0.5)	ND (<0.5)
Citric acid (mg/kg)	ND (<250)	ND (<250)	ND (<250)
Succinic acid (mg/kg)	ND (<1250)	ND (<1250)	ND (<1250)
Adipic acid (mg/kg)	ND (<25)	ND (<25)	ND (<25)
Fumaric acid (mg/kg)	13 ± 0 ^a^	10 ± 0 ^c^	12 ± 0 ^b^
Tartaric acid (mg/kg)	ND (<250)	ND (<250)	ND (<250)
Malic acid (mg/kg)	ND (<500)	ND (<500)	ND (<500)
Lactic acid (mg/kg)	ND (<250)	ND (<250)	ND (<250)
5-Caffeoylquinic acid (mg/kg)	231 ± 11 ^b^	177 ± 9 ^c^	267 ± 1 ^a^
4-Caffeoylquinic acid (mg/kg)	271 ± 3 ^b^	207 ± 2 ^c^	321 ± 6 ^a^
3-Caffeoylquinic acid (mg/kg)	522 ± 7 ^b^	413 ± 14 ^c^	651 ± 13 ^a^
Caffeine (mg/kg)	950 ± 3 ^b^	737 ± 3 ^c^	1152 ± 14 ^a^

The data presented represent mean value ± standard deviation (*n* ≥ 3). Significant differences at the *p* < 0.05 level are denoted by distinct lowercase letters within the same row (a is the largest). “ND” means compound concentration was below the quantification limit of the method, with the legal quantification limit shown in brackets.

**Table 3 foods-14-02840-t003:** VIP values from OPLS-DA models for different cold brew pairs.

Compounds	VIP		
	HPCD and ID	HPCD and TCB	TCB and ID
Phenethyl alcohol	1.21441	-	1.14997
o-Cresol	1.20399	1.23984	−
N-Methylpyrrole-2-carboxaldehyde	−	−	1.08742
Methylcyclopentenolone	1.21466	1.24057	−
m-Cresol	1.15941	−	1.10081
Isophorone	−	1.23505	1.14301
Furfurylideneacetone	1.21428	1.23348	1.14503
Furfuryl alcohol	−	−	1.12274
Furfural	−	−	1.12071
Ethylpyrazine	−	−	1.12331
Difurfuryl ether	1.20627	1.22926	1.11891
6,7-Dihydro-5-methyl-5H-cyclopentapyrazine	1.1496	1.1978	−
5-Methylfurfural	−	−	1.09433
4-Ethyl-2-methoxyphenol	1.18205	1.20819	1.01781
3-Hydroxy-2-methyl-4-pyrone	1.12882	1.17274	1.13325
3-Ethylphenol	1.16704	−	1.12206
3-Ethyl-2,5-dimethylpyrazine	−	−	1.04849
3,5-Diethyl-2-methylpyrazine	1.18088	−	-
2-Thenaldehyde	1.20808	−	1.13853
2-Phenyl-2-butenal	1.21125	1.23993	1.02584
2-Methylpyrazine	−	−	1.13122
2-Methyl-5-propylpyrazine	−	1.22568	1.14657
2-Ethyl-6-methylpyrazine	−	−	1.12093
2-Ethyl-5-methylpyrazine	−	−	1.11541
2-Ethenyl-6-methylpyrazine	1.10941	−	1.14963
2-Acetylpyridine	−	1.2343	1.14187
2-Acetylfuran	−	−	1.11253
2,6-Dimethylpyrazine	−	−	1.12976
2,6-Diethylpyrazine	1.13788	1.17858	-
2,5-Dimethylpyrazine	−	−	1.1256
2,4-Di-tert-butylphenol	1.16794	−	1.1191
2,3-Diethylpyrazine	1.21332	1.23596	−
2,2′-Methylenebisfuran	1.16493	1.24033	1.14866
1-Furfurylpyrrole	1.10399	1.22074	−
1-Acetyl-1,4-dihydropyridine	−	1.23277	1.14171
(±)-Linalool	1.14674	−	1.10164

The minus sign (−) indicates the absence of the compound in a particular extraction process within the combination.

## Data Availability

The original contributions presented in the study are included in the article/Appendix A; further inquiries can be directed to the corresponding authors.

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
