# Peer review of "Comparative Decoding of Physicochemical and Flavor Profiles of Coffee Prepared by High-Pressure Carbon Dioxide, Ice Drip, and Traditional Cold Brew"

_foods, 2025, doi:10.3390/foods14162840_

Round 1

Reviewer 1 Report

Comments and Suggestions for Authors

Dear authors

We appreciate the submission of the manuscript and acknowledge the technical and experimental effort involved. However, following a detailed review, several aspects have been identified that require attention to enhance the clarity, methodological robustness, and practical relevance of the findings. We kindly invite the authors to carefully address the comments and observations provided in the attached document, which highlight both editorial and conceptual elements that should be revised prior to further consideration.

Author Response

We sincerely appreciate your recognition of our technical and experimental effort, as well as your insightful comments and suggestions. We have carefully addressed each point raised in your accompanying document and made substantial revisions to the original manuscript. Below are our detailed responses and modifications corresponding to each valuable comment and suggestion.

Comment 1:The title communicates the general content of the article, but presents lexical redundancy by repeating “cold brew” and lacks technical precision in its structure. It should be reworded as follows

Response: We appreciate the reviewer’s suggestion to enhance the precision of the title.

Accordingly, we have revised the title to Comparative decoding of physicochemical and flavor profiles of coffee prepared by high-pressure carbon dioxide, ice drip, and traditional cold brew.

Hope this revised title more clearly reflects the research objective and contents involved in the study while avoiding repetition.

Comment 2:The abstract is written with informal turns of phrase that detract from academic rigor and lacks a solid conclusion on the applicability of the method evaluated. In addition, the role of statistical analysis to support the results is omitted, which weakens the perception of methodological soundness from the abstract. A poorly focused closure is also observed, without clearly highlighting the practical contribution of the study.

Response: Thank you for pointing out the issues with the abstract. We have revised the abstract accordingly as follows:

High-pressure carbon dioxide (HPCD) has been widely used in the extraction of high-quality bioactive compounds. Flavor profiles of cold brew coffee (CBC) prepared by HPCD, traditional cold brew (TCB), and ice-drip (ID) were comprehensively evaluated by chromatographic approaches, and their variations were investigated by multivariate statistical methods. ID produced the lightest coffee color while HPCD produced the darkest. No significant difference was found in pH among the three coffee processes. The concentrations of chlorogenic acids and caffeine were the highest in ID but the lowest in HPCD. Seventeen of the 48 volatiles were identified as key aroma compounds, contributing nutty, cocoa, caramel, baked, and other coffee flavors to all CBCs. Among them, linalool (OAV=100.50) was found only in ID and provided ID with unique floral and fruity notes; 2-methyl-5-propylpyrazine (OAV=17.70) was found only in TCB and gave roasted aroma. With significantly lower levels of medicine-like and plastic off-flavors, HPCD had a refined aroma experience featuring nutty, cocoa, and caramel notes, though their contents were not the highest. Orthogonal partial least squares-discriminant analysis (OPLS-DA) identified 36 aromas that could differentiate three cold brew methods, with TCB and ID being the most similar. Aroma sensory tests showed that no significant difference was perceived between TCB and HPCD. These findings provide a profound understanding of CBC flavor produced by cold brew methods from the aspect of composition, indicating that HPCD has great potential to realize TCB-like flavor characteristics in a shorter time.

In the revised abstract, the academic writing was improved, statistical differences in key results were specified, and a conclusion on the potential of HPCD in cold brew coffee preparation was highlighted. We believe that these revisions have enhanced the academic quality and clarity of the abstract, ensuring a clear description of the research methods and results.

Comment 3:The introduction presents technical wording problems in phrases such as “CBC is emerging...” and “tons of efforts”, which affects the academic tone. The justification for the use of HPCD is mentioned extensively, but without critically contrasting it with other emerging technologies mentioned, which dilutes the argument. In addition, there is a lack of an explicit formulation of the hypothesis or research question, which limits the structural clarity and focus of the study from the outset.

Response: Thank you for your suggestions on improving the clarity and scholarly rigor of the introduction. We rewrite the whole introduction in Line 31-74.

In the revised introduction, we have revised the technical language and removed informal expressions. The introduction starts with cold brew coffee (CBC). The 2nd paragraph summarizes the major current available techniques (ultrasound-assisted, reduced pressure cycles, and high-pressure processing) to speed up CBC preparation. The 3rd paragraph highlights the superiority of HPCD in bioactive compounds extraction and emphasizes that HPCD had not previously been applied in the field of cold brew coffee. And the final paragraph clearly states the objective of this study.

Comment 4:The 2.1 section is complete but overly detailed; its inventory format affects flow and could be summarized or moved to a table.

Response: Thank you for your valuable suggestions regarding the "Materials and Chemicals" section. We have simplified the details and enhanced the overall flow of this part in Line 77-93, to make ensure only key information is retained.

Comment 5:The 2.2 section clearly describes the procedures, but presents inequality in time and extraction conditions without justifying their comparability, which compromises the methodological fairness of the study.

Response: Thank you for your constructive comments to help improve the quality of this manuscript. The method comparison fairness was carefully considered in this study by balancing the real market situation and comparable coffee extraction yield from different methods.

The traditional cold brew (TCB) procedure and the ice drip (ID) procedure were selected after interviewing baristas from over 20 local coffee shops to reflect common cold brew coffee preparation methods in the real market. The similar TCB procedure was applied in our previous studies [1-2] with solid reproducibility (Line 110-111). The ID procedure is recommended by the manufacturer's instructions for the ice drip coffee apparatus (Line 116-117), which is identical to what is currently used in most coffee shops, and with great sensory repeatability in our preliminary studies.

The coffee yield is a key parameter to evaluate coffee extraction efficiency, since coffee preparation is a procedure to extract coffee solids from beans into the drinking liquids. The HPCD procedures were determined by achieving comparable yields to TCB (~2.3%) and ID (~1.7%) with good operation repeatability. Shown below in the figure is the total soluble solids recovered by 5 MPa HPCD at 20℃ for different periods of time. The results suggested that over 20 min of HPCD treatment can achieve ~2% TDS. To guarantee sufficient extraction, a slightly longer extraction time is recommended. So, 5 MPa, 20℃, 30 min HPCD was selected in this study. In the selected HPCD condition, HPCD yielded 1.73% TDS, which was similar to ID (1.70%) and ready for further analysis.

Reference:

  1. Zhang, D.; Gao, M.; Cai, Y.; Wu, J.; Lao, F. Profiling Flavor Characteristics of Cold Brew Coffee with GC-MS, Electronic Nose and Tongue: Effect of Roasting Degrees and Freeze-Drying. J. Sci. Food Agric. 2024, 104, 6139–6148, doi:10.1002/jsfa.13437.
  2. Cai, Y.; Xu, Z.; Pan, X.; Gao, M.; Wu, M.; Wu, J.; Lao, F. Comparative Profiling of Hot and Cold Brew Coffee Flavor Using Chromatographic and Sensory Approaches. Foods 2022, 11, 2968, doi:10.3390/foods11192968.

Comment 6:Although the 2.3.7 section rigorously describes the analytical procedures, a methodological justification explaining the choice of certain critical parameters (e.g., extraction times, chromatographic columns, standards) is lacking. Furthermore, neither the detection limits nor the validation of the applied methods, key aspects to guarantee the robustness and reproducibility of the results, are discussed.

Response: Thank you for your valuable comments.

We have modified the method section as suggested, to specify the extraction time (Line 194-197), chromatographic column information (Line 199-200), and internal standard (Line 192). The aroma GC separation and volatiles MS identification are well-validated classical protocols (Line 202-210) that have been applied in various coffee research [1-6], and only aroma with matching quality > 85% was identified, showing satisfactory robustness and reproducibility of the results. In addition, the aroma retention time, MS data, and contents in this study were overall aligned with our previous cold brew coffee research, suggesting the method is working well. Thus, we reasonably believe that the method in 2.3.7 section was solid to support our results and conclusion.

Reference:

  1. Zhang, D.; Gao, M.; Cai, Y.; Wu, J.; Lao, F. Profiling Flavor Characteristics of Cold Brew Coffee with GC-MS, Electronic Nose and Tongue: Effect of Roasting Degrees and Freeze-Drying. J. Sci. Food Agric. 2024, 104, 6139–6148, doi:10.1002/jsfa.13437.
  2. Cai, Y.; Xu, Z.; Pan, X.; Gao, M.; Wu, M.; Wu, J.; Lao, F. Comparative Profiling of Hot and Cold Brew Coffee Flavor Using Chromatographic and Sensory Approaches. Foods 2022, 11, 2968, doi:10.3390/foods11192968.
  3. Heo, J.; Adhikari, K.; Choi, K.S.; Lee, J. Analysis of Caffeine, Chlorogenic Acid, Trigonelline, and Volatile Compounds in Cold Brew Coffee Using High-Performance Liquid Chromatography and Solid-Phase Microextraction—Gas Chromatography-Mass Spectrometry. Foods 2020, 9, 1746, doi:10.3390/foods9121746.
  4. Piccino, S.; Boulanger, R.; Descroix, F.; Sing, A.S.C. Aromatic Composition and Potent Odorants of the “Specialty Coffee” Brew “Bourbon Pointu” Correlated to Its Three Trade Classifications. Food Res. Int. 2014, 61, 264–271, doi:10.1016/j.foodres.2013.07.034.
  5. Amanpour, A.; Selli, S. Differentiation of Volatile Profiles and Odor Activity Values of Turkish Coffee and French Press Coffee. J. Food Process. Preserv. 2016, 40, 1116–1124, doi:10.1111/jfpp.12692.
  6. Caporaso, N.; Whitworth, M.B.; Cui, C.; Fisk, I.D. Variability of Single Bean Coffee Volatile Compounds of Arabica and Robusta Roasted Coffees Analysed by SPME-GC-MS. Food Res. Int. 2018, 108, 628–640, doi:10.1016/j.foodres.2018.03.077.

Comment 7:The model validation criteria beyond the use of Q² and R² are not detailed, nor is it mentioned whether basic statistical assumptions were checked, which limits the methodological transparency and reproducibility of the analysis.

Response: We appreciate your valuable comment on our statistical analysis.

For the OPLS-DA model, besides Q² and R², the permutation test for 200 times was also applied to check if the model was overfitted or not (Line 478-482). Table S2 in the supplementary data provides more detailed information on the model validation outputs. Our OPLS-DA model passed all the above-mentioned tests. This model validation procedure was also employed in other studies to verify the model accuracy and reliability [1-9]. Moreover, basic statistical assumptions, such as the equal variance assumption, were checked when running one-way ANOVA (screenshot is shown below). And all data passed the basic statistical assumptions checking. We thus believe that the model we have is solid enough for data interpretation.

Reference:

  1. Hou, H.; Tang, Y.; Zhao, J.; Debrah, A.A.; Shen, Z.; Li, C.; Du, Z. Authentication of Organically Produced Cow Milk by Fatty Acid Profile Combined with Chemometrics: A Case Study in China. J. Food Compos. Anal. 2023, 120, 105297, doi:10.1016/j.jfca.2023.105297.
  2. Hu, D.; Yang, G.; Liu, X.; Qin, Y.; Zhang, F.; Sun, Z.; Wang, X. Comparison of Different Drying Technologies for Coffee Pulp Tea: Changes in Color, Taste, Bioactive and Aroma Components. LWT 2024, 200, 116193, doi:10.1016/j.lwt.2024.116193.
  3. Yan, J.; Chen, J.; Huang, Z.; He, L.; Wu, L.; Yu, L.; Zhu, W. Characterisation of the Volatile Compounds in Nine Varieties and Three Breeding Selections of Celery Using GC–IMS and GC–MS. Food Chem.: X 2024, 24, 101936, doi:10.1016/j.fochx.2024.101936.
  4. Yang, X.; Chen, Q.; Liu, S.; Hong, P.; Zhou, C.; Zhong, S. Characterization of the Effect of Different Cooking Methods on Volatile Compounds in Fish Cakes Using a Combination of GC–MS and GC-IMS. Food Chem.: X 2024, 22, 101291, doi:10.1016/j.fochx.2024.101291.
  5. Zhang, J.; Zhong, L.; Wang, P.; Song, J.; Shi, C.; Li, Y.; Oyom, W.; Zhang, H.; Zhu, Y.; Wen, P. HS-SPME-GC-MS Combined with Orthogonal Partial Least Squares Identification to Analyze the Effect of LPL on Yak Milk’s Flavor under Different Storage Temperatures and Times. Foods 2024, 13, 342, doi:10.3390/foods13020342.
  6. Zhang, W.; Qian, S.; Wu, D.; Yan, Q.; Chung, J.-P.; Jiang, Y. Dynamic Environmental Interactions Shape the Volatile Compounds of Agarwood Oils Extracted from Aquilaria Sinensis Using Supercritical Carbon Dioxide. Molecules 2025, 30, 945, doi:10.3390/molecules30040945.
  7. Zhang, G.; Liu, Y.; Luo, Y.; Zhang, C.; Li, S.; Zheng, H.; Jiang, X.; Hu, F. Comparison of the Physicochemical Properties, Microbial Communities, and Hydrocarbon Composition of Honeys Produced by Different Apis Species. Foods 2024, 13, 3753, doi:10.3390/foods13233753.
  8. Xie, L.; Guo, S.; Rao, H.; Lan, B.; Zheng, B.; Zhang, N. Characterization of Volatile Flavor Compounds and Aroma Active Components in Button Mushroom (Agaricus Bisporus) across Various Cooking Methods. Foods 2024, 13, 685, doi:10.3390/foods13050685.
  9. Li, K.; Zhang, L.; Yi, D.; Luo, Y.; Zheng, C.; Wu, Y. Insights into the Volatile Flavor Profiles of Two Types of Beef Tallow via Electronic Nose and Gas Chromatography–Ion Mobility Spectrometry Analysis. Foods 2024, 13, 1489, doi:10.3390/foods13101489.

Comment 8:The 3.1 section presents a solid comparative analysis, but lacks a critical discussion of the sensory relevance of differences in the reported nonvolatile compounds. In addition, HPCD efficiency is interpreted in terms of time without considering possible impacts on organoleptic perception or consumer acceptance, which limits the depth of the applied analysis.

Response: We appreciate the very professional comment on the non-volatile compounds analysis.

The compounds we quantified are all relevant to certain kinds of coffee sensory experience. For example, sugars are relevant to sweetness, acids are relevant to sourness, caffeine is relevant to bitterness, and CQAs are relevant to bitterness and astringency. As suggested, we added these taste relevancies into our contents of compounds discussion (Line 292-319). However, the HPCD facility we have is not food grade, so it was not recommended/allowed to deal with samples for human consumption. In compensation, we ran an aroma sensory test (Line 218-230, 389-405), asking our panelists to smell only, to evaluate the possible impacts on olfactory perception or consumer acceptance. We hope this revision helps improve the depth of the discussion.

Comment 9:The 3.2 section comprehensively identifies volatile compounds and their OAVs, but lacks a critical linkage to actual consumer sensory perception. In addition, the aromatic profile is described based on statistical differences without contextualizing its practical relevance or impact on the sensory quality of the final product.

Response: We appreciated your suggestion to improve the quality of the aroma data interpretation. As suggested, we supplemented an aroma sensory test to link to actual consumer sensory perception (Line 218-230). And a Pearson correlation analysis was conducted between the sensory scores and volatile compounds with OAV>1 (Line 237-238) to reveal the practical impact on the aroma quality of the final product. The related supplementary content has been added in Section 3.2.2 Aroma Sensory Analysis Based on Odor Active Compounds (Line 389-439), to contextualize the potential dose-effect relationship of volatile compounds from a practical perspective.

Comment 10:Although the Figure 4 heat map facilitates comparative visualization, the interpretation of the grouping is superficial and does not discuss the chemical or sensory significance of the groups formed. The opportunity to link the clustering patterns to relevant functional or technological characteristics of the product is missed. Please improve Figure 4.

Response: Thank you for the very constructive suggestions regarding hierarchical clustering heatmaps.

As suggested, we reanalyzed the OPLS-DA to get the 36 differential volatile compounds among the three methods, from which we generated the improved figure showing the grouping of the three methods with the hierarchical clustering analysis. Based on the new heatmap, we supplemented chemical or sensory significance of the groups formed (Line 490-500), and the clustering patterns to relevant sensory and technological characteristics of the coffee product were added (Line 501-510). Hope this revision can fulfill your expectations on interpreting HCA information.

Comment 11:The conclusions adequately summarize the findings, but present an affirmative tone that does not reflect the methodological limitations of the study, such as the lack of sensory validation or consumer acceptance studies.

Response: Thank you for your thoughtful suggestion.

As suggested, the conclusion was adjusted by supplementing the research limitations in the end (Line 540-542). 

Reviewer 2 Report

Comments and Suggestions for Authors

The study evaluates the impact of different techniques on certain parameters of cold brewed coffee. The experimental design is correct, however, it feels that statistical may not be the most suitable. 

What was the % of match with volatile compounds with NIST library

Why this the authors chose OPLS-DA. With such a small number of samples, PCA as an unsupervised method would be a better used as a starting exploratory tool . The use of DA is overfitting the model forcing differences between samples. 

If doing OPLS-DA the use of S plots would help identify unique variables associated to each class

Similarly for PLS, what are the variables to compare, what are X and Y variables, specify. What is your predictor variable. 

Author Response

We sincerely appreciate the reviewers' recognition of the experimental design and valuable suggestions regarding the statistical methods employed. In accordance with your comments, we have made corresponding revisions to the statistical analysis section of the manuscript, as detailed below.

Comment 1: What was the % of match with volatile compounds with NIST library?

Response: We sincerely appreciate your pointing this out, as it is crucial for the identification of volatile compounds.

We have specified this information in Line 206-208: Volatile compounds were identified by comparing mass spectra to the standard NIST17 library, and the compounds with a matching quality over 85% were identified.

Comment 2: Why this the authors chose OPLS-DA. With such a small number of samples, PCA as an unsupervised method would be a better used as a starting exploratory tool. The use of DA is overfitting the model forcing differences between samples. 

Response: We sincerely appreciate the reviewers' professional suggestions regarding the use of PCA as an initial tool for exploratory analysis.

This has been incorporated in our revised manuscript; the differentiation of compound identification was started with the unsupervised PCA, to make sure the three methods can be well separated (Line 443-465).

In response to the risk of overfitting in the OPLS-DA model, fitting index (R²), predictive ability index (Q²), and 200 times permutation tests were applied to evaluate whether the model was overfitted or not (Line 478-482). Table S2 in the supplementary data provides more detailed information on the model validation outputs. Our OPLS-DA model passed all the above-mentioned tests. This model validation procedure was also employed in other studies to verify the model accuracy and reliability [1-9]. We thus reasonably believe that the model we have is solid enough for data interpretation.

To be more specific regarding the model overfitting check, each data point in the permutation test graph corresponds to the outcome of a model training session: the green circles represent the fitting index (R²), and the blue squares represent the predictive ability index (Q²). The two points on the far right show the original R² and Q² values of the model, while the dense cluster of points on the left illustrates the distribution of R² and Q² values from models rebuilt 200 times after randomly permuting the class labels. After 200 permutation tests, if the original R² and Q² values on the right side of the graph are higher than those on the left, and the regression line of the blue Q² points intersects the vertical axis at or below zero (i.e., the Y-intercept of the Q² regression line is less than zero), it indicates that the model is not overfitted and that the results are reliable. Conversely, if the Q² intercept is greater than zero, it suggests model overfitting. In our study, all models underwent repeated permutation tests, and none showed signs of overfitting.

Reference:

  1. Hou, H.; Tang, Y.; Zhao, J.; Debrah, A.A.; Shen, Z.; Li, C.; Du, Z. Authentication of Organically Produced Cow Milk by Fatty Acid Profile Combined with Chemometrics: A Case Study in China. J. Food Compos. Anal. 2023, 120, 105297, doi:10.1016/j.jfca.2023.105297.
  2. Hu, D.; Yang, G.; Liu, X.; Qin, Y.; Zhang, F.; Sun, Z.; Wang, X. Comparison of Different Drying Technologies for Coffee Pulp Tea: Changes in Color, Taste, Bioactive and Aroma Components. LWT 2024, 200, 116193, doi:10.1016/j.lwt.2024.116193.
  3. Yan, J.; Chen, J.; Huang, Z.; He, L.; Wu, L.; Yu, L.; Zhu, W. Characterisation of the Volatile Compounds in Nine Varieties and Three Breeding Selections of Celery Using GC–IMS and GC–MS. Food Chem.: X 2024, 24, 101936, doi:10.1016/j.fochx.2024.101936.
  4. Yang, X.; Chen, Q.; Liu, S.; Hong, P.; Zhou, C.; Zhong, S. Characterization of the Effect of Different Cooking Methods on Volatile Compounds in Fish Cakes Using a Combination of GC–MS and GC-IMS. Food Chem.: X 2024, 22, 101291, doi:10.1016/j.fochx.2024.101291.
  5. Zhang, J.; Zhong, L.; Wang, P.; Song, J.; Shi, C.; Li, Y.; Oyom, W.; Zhang, H.; Zhu, Y.; Wen, P. HS-SPME-GC-MS Combined with Orthogonal Partial Least Squares Identification to Analyze the Effect of LPL on Yak Milk’s Flavor under Different Storage Temperatures and Times. Foods 2024, 13, 342, doi:10.3390/foods13020342.
  6. Zhang, W.; Qian, S.; Wu, D.; Yan, Q.; Chung, J.-P.; Jiang, Y. Dynamic Environmental Interactions Shape the Volatile Compounds of Agarwood Oils Extracted from Aquilaria Sinensis Using Supercritical Carbon Dioxide. Molecules 2025, 30, 945, doi:10.3390/molecules30040945.
  7. Zhang, G.; Liu, Y.; Luo, Y.; Zhang, C.; Li, S.; Zheng, H.; Jiang, X.; Hu, F. Comparison of the Physicochemical Properties, Microbial Communities, and Hydrocarbon Composition of Honeys Produced by Different Apis Species. Foods 2024, 13, 3753, doi:10.3390/foods13233753.
  8. Xie, L.; Guo, S.; Rao, H.; Lan, B.; Zheng, B.; Zhang, N. Characterization of Volatile Flavor Compounds and Aroma Active Components in Button Mushroom (Agaricus Bisporus) across Various Cooking Methods. Foods 2024, 13, 685, doi:10.3390/foods13050685.
  9. Li, K.; Zhang, L.; Yi, D.; Luo, Y.; Zheng, C.; Wu, Y. Insights into the Volatile Flavor Profiles of Two Types of Beef Tallow via Electronic Nose and Gas Chromatography–Ion Mobility Spectrometry Analysis. Foods 2024, 13, 1489, doi:10.3390/foods13101489.

Comment 3: If doing OPLS-DA, the use of S plots would help identify unique variables associated with each class.

Response: We sincerely appreciate the reviewer’s suggestion to include S-plots for identifying unique variables associated with each class. As suggested, S-plots were introduced to identify unique variables associated to each cold brew method (Figure 6), and related discussion was supplemented (Line 483-500).

Comment 4: Similarly, for PLS, what are the variables to compare, what are the X and Y variables, specify. What is your predictor variable. 

We sincerely thank you for pointing out the issues with the PLS-DA section.

The objective of this study is to decode the coffee flavor profile variation among three cold brew methods from a compositional aspect. Prediction is not involved in this study. Thus, we deleted the whole PLS-DA section, to enhance the manuscript's clarity and make the whole structure more straightforward.

We believe your valuable suggestions have helped the research focus more sharply on key findings regarding the impact of different coffee extraction methods on composition, significantly improving the study's academic quality.

Round 2

Reviewer 1 Report

Comments and Suggestions for Authors

Dear authors

Many thanks for your response.

Regards